# Impacts of Pollutant Emissions from Typical Petrochemical Enterprises on Air Quality in the North China Plain

**Ziyue Zhang [1,*], Wenyu Yang [1], Shucai Zhang [1] and Long Chen [2,*]**

1　State Key Laboratory of Safety and Control for Chemicals, SINOPEC Research Institute of Safety Engineering Co., Ltd., Qingdao 266000, China

2　Key Laboratory of Geographic Information Science (Ministry of Education), School of Geographic Sciences, East China Normal University, Shanghai 200241, China

*　Correspondence: zzy.qday@sinopec.com (Z.Z.); chenlong@geo.ecnu.edu.cn (L.C.); Tel.: +86-0532-8378-6397 (Z.Z.)

**Abstract:** Under the state's key surveillance, petrochemical industries are considered polluting enterprises. Even though large-scale petrochemical enterprises follow the complete treatment of combustion waste gas, process waste gas, and volatile organic waste gas pollutants, the impact of pollutant emissions on the regional air quality is unclear. This study used the atmospheric chemical transport model and adopted the subtraction method to simulate the impacts of air pollutant emissions from four typical petrochemical enterprises on regional air quality of the North China Plain. Results indicated that emissions from petrochemical enterprises on surface $PM_{2.5}$, $SO_2$, and $NO_2$ concentrations mainly contributed to the nearby area, particularly $SO_2$ and $NO_2$. The pollution can be controlled within the boundaries of the petrochemical plants. Petrochemical enterprises had a small $SO_2$ and $NO_2$ contribution with a maximum of up to 4.65% within a 9 km distance. Emissions from petrochemical enterprises contributed less to surface $PM_{2.5}$ concentrations (less than 0.5%) within a 9 km distance. Surface $O_3$ concentrations driven by petrochemical enterprises did not show near-source distribution characteristics, which were closely related to its complex precursors and secondary reactions. Contributions of petrochemical enterprises to local pollution decreased significantly with the increase in distance. The $SO_2$ and $NO_2$ pollution contributions to the North China Plain remained around 0.1–0.2%, with the maximum contribution occurring in January and July. The maximum contribution of $PM_{2.5}$ in this region was in April (0.42%) while it was below 0.1% for other months. The pollutant emission from the four typical petrochemical enterprises in the North China Plain had little impact on the concentration of air pollutants in the North China Plain. However, it had a significant impact on the ambient air quality in the region near the enterprise. This study can be useful in analyzing and refining the influence of enterprises on the region.

**Keywords:** petrochemical enterprise; atmospheric pollution; WRF-GC model; the North China Plain

## 1. Introduction

With the increase in industrialization and urbanization, air pollution incidents have been frequently occurring in China. Air pollution issues caused by $PM_{2.5}$ and $O_3$ have become increasingly prominent. Cities in the North China Plain nearly occupied the bottom 20 places in China's air quality rankings [1]. Air pollution has drawn extensive attention from the government and general public as one of the greatest environmental and public health threats. To improve the air quality, the Ministry of Ecological Environment of China announced a new standard for ambient air quality, which includes six criterion pollutants: sulfur dioxide ($SO_2$), nitrogen dioxide ($NO_2$), carbon monoxide ($CO$), ozone ($O_3$), fine particulate matter ($PM_{2.5}$), and inhalable particulate matter ($PM_{10}$). The air pollution condition in China has significantly improved compared with the condition in 2014 thanks to the 'Air Pollution Prevention and Control Action Plan' promulgated in 2013 and the

'Three-year Action Plan to Win the Blue Sky Defense War' in 2018 [2]. However, 43.3% of ambient air quality in cities has not met the standard yet. Annual air pollution days in the North China Plain accounted for 63.5%. The $PM_{2.5}$, $O_3$, $PM_{10}$, and $NO_2$ as the primary pollutant accounted for 48.0%, 46.6%, 5.3%, and 0.2% of the total pollution days, respectively [3].

In general, changing land use and introducing additional anthropogenic emissions are two pathways to have an impact on air quality [4,5]. Even though industrial enterprises generate huge economic benefits and are considered an important pillar of economic development in China, they also lead to large-scale air pollution. Various pollutants including sulfur oxides ($SO_x$), nitrogen oxides ($NO_x$), particulate matter, volatile organic compounds (VOCs), $NH_3$, CO, $H_2S$, and trace metals are discharged during various phases of the petroleum refining process [6]. The emissions of $SO_2$, $NO_x$, and VOCs from the petroleum and chemical industry accounted for 14%, 10%, and 42% of total emissions in the industrial sector in 2015, respectively [7]. Numerous studies have illustrated the significant impact of the petrochemical industry on $PM_{2.5}$ pollution [8–10].

It is extremely important to reduce emissions from petrochemical industries. The Chinese government announced ambitious goals in 2017 to build a beautiful China, reverse the deteriorating trend of the ecological environment from the source, and contribute to global ecological security [11]. As the industry is the primary source of emissions, reducing industry emissions is crucial to achieving these goals. The Ministry of Ecological Environment of China and associated petrochemical enterprises have made considerable efforts to reduce pollution in the refining industry. Sinopec is one of the top 500 global companies and the biggest oil refiner in China. Sinopec has spent tens of billions of CNY to carry out special governance actions while focusing on total pollutant emission reduction, standard upgradation and transformation, detection and control of volatile organic contaminants, odor control, and environmental risk prevention [12]. Although previous studies have analyzed the impact of industrial pollution on the atmospheric environment [7,9], few studies were related to the impact of emissions from refining and chemical enterprises on the air quality of cities or urban agglomerations. Considering the removal of heavily polluting enterprises from Beijing and the constantly improving standard of enterprises, the impact of the petrochemical refining industry on regional air quality needs further evaluation.

In this study, an atmospheric chemical transport model was used to investigate the impact of typical petrochemical enterprises on air quality in the North China Plain. The model was based on the real-time monitoring of pollutant emissions from petrochemical enterprises. This study is crucial for accurately determining the impact of current pollutant emissions from typical petrochemical enterprises, formulating and implementing targeted environmental protection plans, and responding to the deployment of pollution prevention and control.

## 2. Materials and Methods

### 2.1. Study Domain

The North China Plain (NCP) and some eastern, central, northwestern, and northeastern regions in China were included in the study domain, which was from 105° E to 127° E and from 30° N to 45° N. The locations of four studied petrochemical enterprises are shown in Figure 1. Enterprises C and D are located in Hebei while enterprises A and B are located in Tianjin and Beijing, respectively. Enterprises A covers an area of 16.4 square kilometers, which owns 34 sets of refining units, 21 sets of chemical units, and a comprehensive supporting crude oil processing capacity of 12.5 million tons/year. Enterprise B covers an area of 37.6 square kilometers with 88 sets of production units, 71 sets of auxiliary production units, and 10 million tons of oil refining capacity. Enterprise C covers an area of 1.8 square kilometers with 20 main units and a refining capacity of 3.5 million tons/year. Enterprise D covers an area of 0.6 square kilometers with 26 refining units and a crude oil processing capacity of 8 million tons/year.

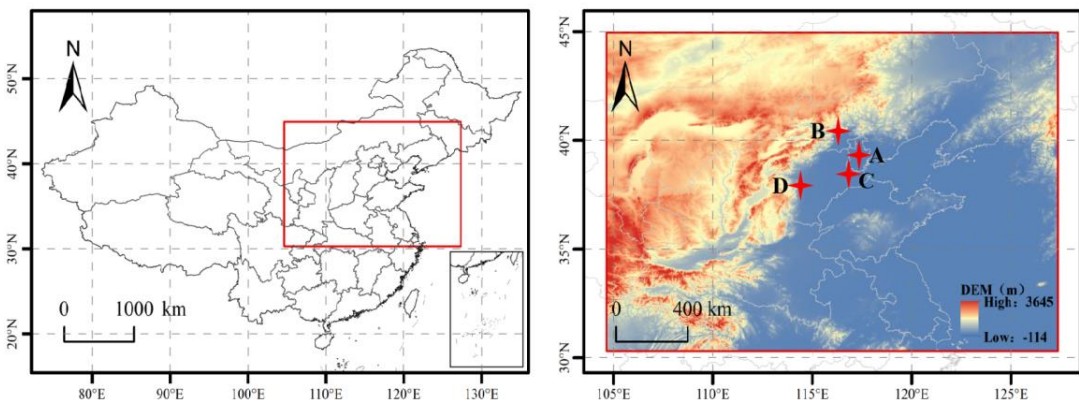

**Figure 1.** Study domain and terrain height in the domain.

## 2.2. Preprocessing of Field Measurement Data

### 2.2.1. Acquisition and Processing of Industrial Pollutant Emission Data

The emission data used in this study were obtained from continuous emission monitoring systems (CEMSs) of the four enterprises. This study improved the data quality through the processing of abnormal and missing data. Although CEMS has existed for decades, it suffers calibration and data quality issues caused by factors such as mechanical problems and network failure [13]. A high-dimensional random matrix model was constructed to detect abnormal data during data processing. The abnormal data must be revised according to the ecological environment standard of China, "Specifications for continuous emissions monitoring of $SO_2$, $NO_x$, and particulate matter in the flue gas emitted from stationary sources (HJ 75-2017)" (hereafter referred to as the CEMS standard). We followed section 12.2 of the CEMS standard to remove the complete record and fill in the missing data.

### 2.2.2. Normalization of Data

In order to eliminate the dimensional influence between different evaluation indexes, we normalized the data to solve the problem that the data indexes of different dimensions are not comparable. We used the min–max standardized processing method to normalize the air quality data, meteorological data, and industrial pollutant emission data. The processed data values were between 0 and 1.

## 2.3. Model Description

### 2.3.1. Atmospheric Chemical Transport Model

WRF-GC (version 1.0; https://fugroup.org/index.php/WRF-GC, accessed on 18 January 2023) is an online two-way combination of the Weather Research and Forecasting (WRF) meteorological model and the chemical transport model (GEOS-Chem) [14,15]. WRF-GC inherits the state-of-art representatives of meteorological processes and atmospheric chemical processes. The GEOS-Chem model is a global 3D chemical transport model which includes detailed atmospheric chemical mechanisms and various aerosol processes [16–19]. The model includes fully coupled ozone-$NO_x$-VOC-halogen-aerosol chemistry and can simulate the chemical process of most trace chemical components in the atmosphere, including ozone, particulate matter, nitrogen oxides, and sulfur compounds. The coupled physicochemical processes include emission, transport, deposition, chemical transformation, photolysis, and aerosol processes of trace chemical components.

The WRF model is a powerful tool to simulate atmospheric physical processes. It was used to reproduce the airflows over the surface of Lakes [20] and the atmospheric circulation during the summer season in a coastal region of central Italy [21]. The model was also used to estimate the impacts of Earth surface conditions on the accurate prediction of air temperature in Belarus [22]. Previous studies have provided the evidence of good application in mesoscale simulation for the model.

Model simulations were conducted with the horizontal resolution of 9 km $\times$ 9 km and 47 vertical hybrid eta levels from the surface to 0.01 hPa. The basic meteorological data of WRF adopted the 6 h time-series data set of the FNL reanalysis field of the National Centers for Environmental Prediction (NCEP). Physical options used in our WRF-GC simulation includes Morrison two-moment scheme [23] for microphysics, RRTMG scheme [24] for longwave and shortwave radiation, MM5 Monin–Obukhov scheme [25] for surface layer, Noah scheme [26,27] for land surface, MYNN2 scheme [28] for planetary boundary layer, and New Tiedtke scheme [29] for cumulus. Four study periods (first weeks in January, April, July, and October) in 2019 were chosen to represent the different pollution conditions in each season of the year including two heavy pollution events (first week in October and July) in the NCP. Simulations in the anterior four days of each week were dedicated to spin-up while not being involved in analysis and simulations in the posterior three days were dedicated to analysis.

2.3.2. Model Inputs and Evaluation

Monthly anthropogenic emissions in the domain were obtained from the Multi-resolution Emission Inventory for China (MEIC) [30]. The emissions of atmospheric pollutants (i.e., $SO_2$, $NO_x$, and VOCs) for the four typical petrochemical enterprises observed from CEMS were used as point sources in this model. Two model scenarios were conducted, including a base case scenario with only MEIC emissions and a sensitivity scenario with the sum of MEIC emissions and emissions from petrochemical enterprises. The difference between the two scenarios indicated the contributions of pollutant emissions from the four typical petrochemical enterprises to air quality in the study domain.

Observations derived from the Chinese national air quality monitoring network (China National Environmental Monitoring Center, http://106.37.208.233:20035 (accessed on 26 August 2020)) were used for model evaluation. The observed data include the daily average of $PM_{2.5}$, $SO_2$, and $NO_2$ atmospheric concentrations with a time-frequency of 1 h. A total of 760 monitoring stations were located in the study domain, and we integrated them into 238 cities. The cities with five or more monitoring stations were involved in the evaluation. Eventually, a total of 50 cities were involved and the average of observations in each city was archived. The simulation ability of the model was evaluated through the scatter diagram and the relationship between the observed and simulated values was determined.

## 3. Results

### 3.1. Emissions of Atmospheric Pollutants from Typical Petrochemical Enterprises

The total annual emissions of $SO_2$, $NO_x$, and VOCs from the four typical petrochemical enterprises in 2019 were obtained after cleaning and analyzing online monitoring data (Table 1). The total emissions of $SO_2$ and $NO_x$ were more than 2000 tons/year, whereas the total emissions of $NO_x$ and VOCs were more than 2000 tons/year. The annual emission of pollutants is mainly related to the scale of enterprises. Enterprises A and B are larger in scale and discharge more pollutants.

**Table 1.** $SO_2$ and $NO_x$ emissions from four typical petrochemical enterprises in 2019.

| Enterprises | Annual $SO_2$ Emission (t/a) | Annual $NO_x$ Emission (t/a) | Annual VOCs Emission (t/a) |
|---|---|---|---|
| A | 235.5 | 1146.1 | 2104.3 |
| B | 59.4 | 1003.6 | 1968.0 |
| C | 42.9 | 125.9 | 435.8 |
| D | 77.6 | 216.0 | 763.0 |
| Total | 415.3 | 2491.7 | 5271.2 |

Waste gas pollution sources are mainly divided into organized emission sources and unorganized emission sources according to the type of emission. The unorganized waste gas mainly generates from process unit areas, petrochemical tank farms, pipelines, and

sewage treatment plants. There are various degrees of leakage and dissipation from the disorganized waste gas and the emission is little affected by the production situation. Therefore, the annual VOC emissions from the four petrochemical enterprises were relatively stable without large fluctuations (Figure 2).

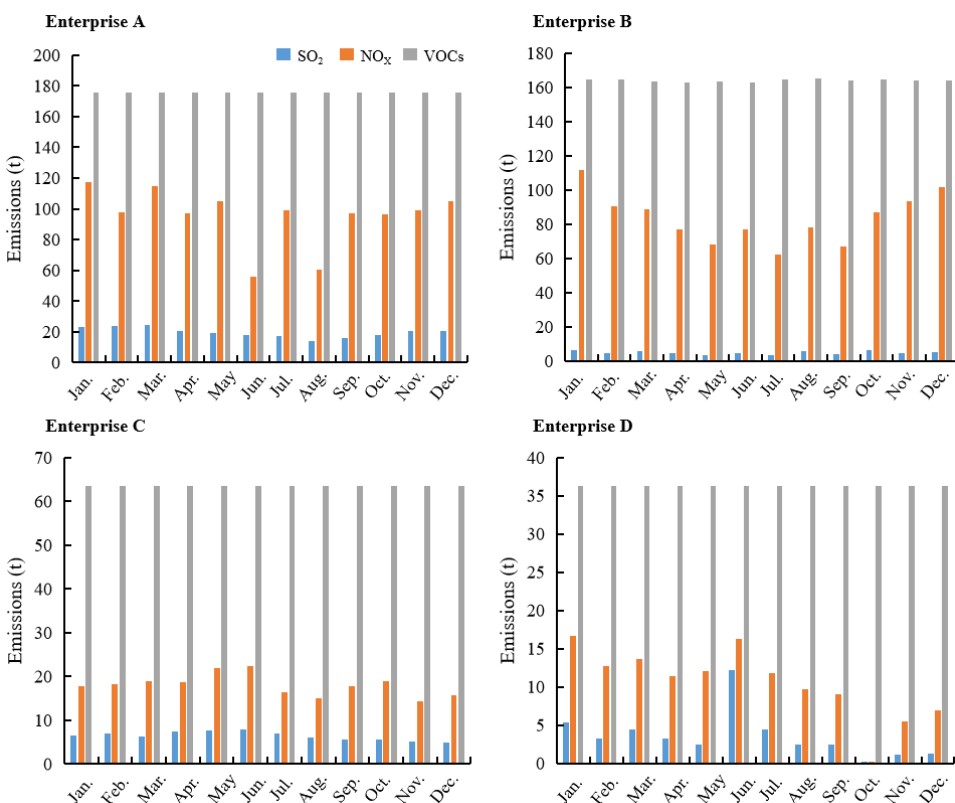

**Figure 2.** Seasonal emissions of main air pollutants from the four petrochemical enterprises.

The organized waste gas sources mainly include heating furnace flue gas, catalytic coking flue gas, boiler flue gas, and ethylene device. These sources are closely related to the production status. The seasonality of $SO_2$ emissions from enterprises A, B, and C was weak. However, strong seasonality was observed for the $SO_2$ emission from enterprise D. Emissions in June were several times higher than that in other months, whereas emissions in October were considerably low. The overall emissions in the first half of the year were higher than those in the second half of the year. Strong seasonality was observed for the emission of $NO_x$ from enterprises A and B with lower emissions in summer. Weak seasonality was observed for enterprise C and a gradual decrease in the emissions was observed for enterprise D.

### 3.2. Model Evaluation and Spatial-Temporal Distribution of Air Pollutant Concentrations

3.2.1. Evaluation of Model Performance

The simulated concentrations of $PM_{2.5}$, $O_3$, $SO_2$, and $NO_2$ from the four months in 2019 were used to evaluate our model performance (Figure 3). The scatter patterns illustrated credible performance for the simulations of $PM_{2.5}$, $SO_2$, and $NO_2$ concentrations. The ratios of simulated values to observed values were primarily scattered near the 1:1 solid line (Figure 3A,C,D). Meanwhile, low root mean squared error (RMSE) also illustrated high aggregation degree of scatters. The model performance on the maximum daily 8 h average (MDA8) $O_3$ was slightly worse than that of the other three types of pollutants. The simulated values to some extent underestimated the observed values (Figure 3B).

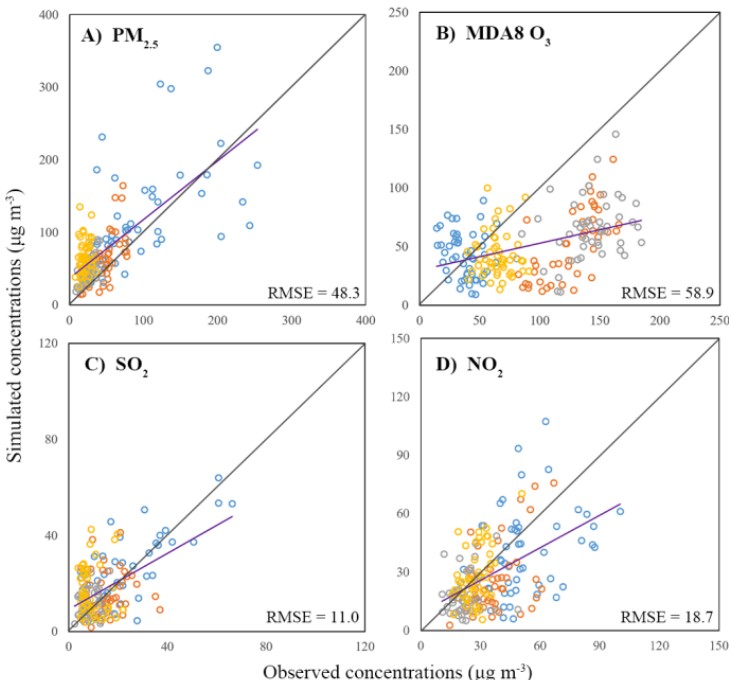

**Figure 3.** Comparison between observed concentrations and simulated concentrations of (**A**) PM$_{2.5}$, (**B**) MDA8 O$_3$, (**C**) SO$_2$ and (**D**) NO$_2$ air pollutants in 50 cities located in the study domain. The blue circles, orange circles, gray circles, and yellow circles indicate air pollutant concentrations in January, April, July, and October, respectively. The middle solid line represents the 1:1 line and the purple solid lines represent the trend of datasets. RMSE represents root mean squared error (unit: μg m$^{-3}$).

Indeed, the monitoring site was located in a given site while the model results represented the average conditions in a region with coarse horizontal resolution (9 km × 9 km) and coarse vertical stratification (altitude of surface layer, 123 m). Owing to the representation of average conditions, the model may not capture the local conditions around the monitoring site adequately, such as point emission source, local microclimate, and vertical thermal structure [21]. These discrepancies particularly contributed to model bias in air pollutants such as O$_3$ which have complex chemical processes with large inputs of meteorological variables. Higher horizontal and vertical resolution with more accurate meteorological and emission inputs are needed in future studies to improve regional simulations. Nevertheless, the impacts of pollutant emissions on air quality in this study were illustrated as percentage contribution, and the bias may be reduced because bias from different scenarios may largely offset each other with normalization.

### 3.2.2. Spatial-Temporal Distribution of Air Pollutant Concentrations

Figure 4 shows the spatial-temporal distribution of each air pollutant concentration (PM$_{2.5}$, MDA8 O$_3$, SO$_2$, and NO$_2$) over the NCP in 2019. High concentrations of PM$_{2.5}$, SO$_2$, and NO$_2$ appeared in winter (January), while low concentrations appeared in summer (July). In winter, the concentrations hotspots were distributed in northern China. The hotspots for the three pollutants did not completely overlap, but the PM$_{2.5}$ hotspots were widespread and reached as far as central China. The SO$_2$ and NO$_2$ high-concentration areas were limited. Due to the short atmospheric lifetime, both of them were easily transformed and removed, resulting in the near-source distribution of these two pollutants.

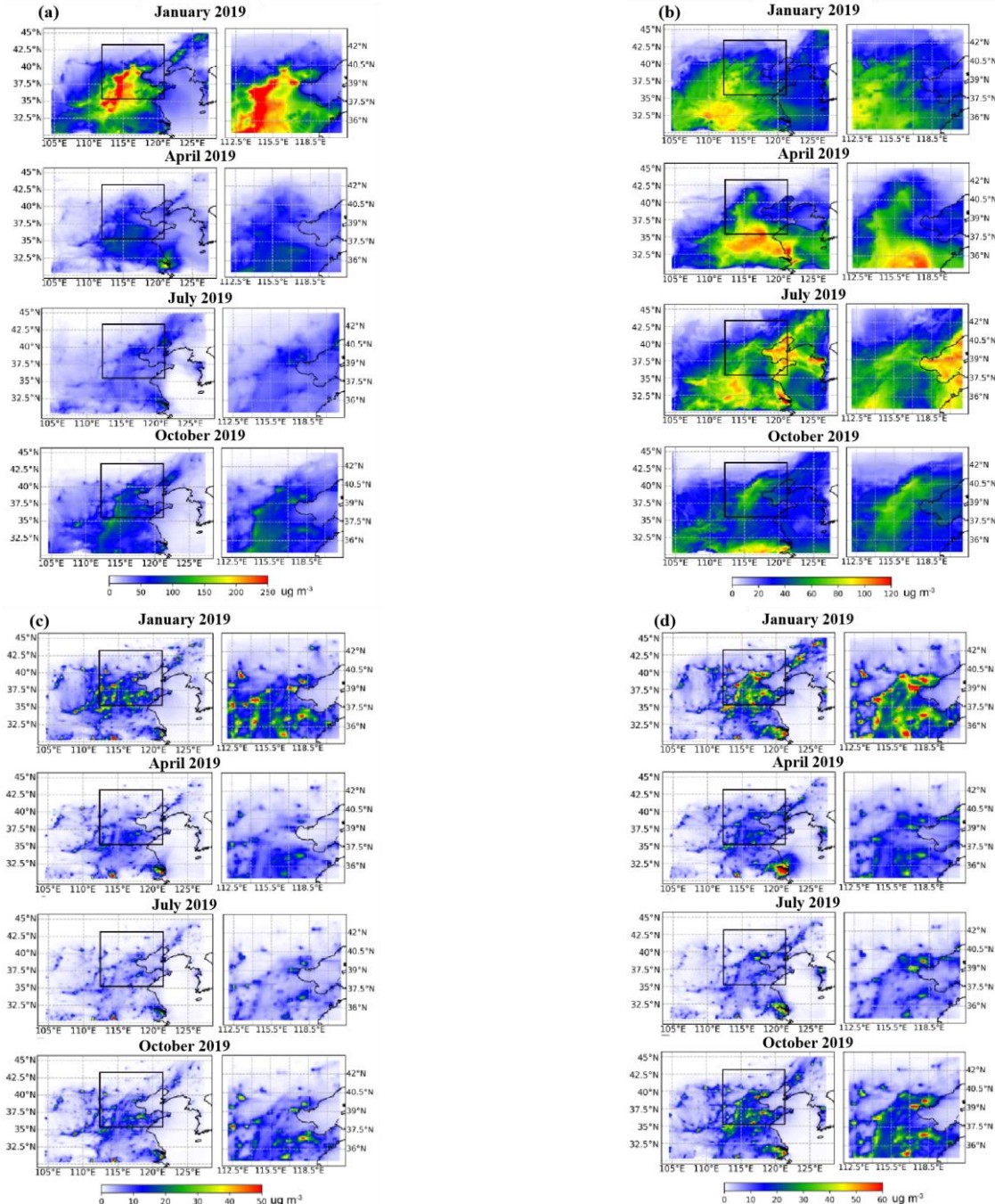

**Figure 4.** Spatial distribution of surface (**a**) PM$_{2.5}$, (**b**) MDA8 O$_3$, (**c**) SO$_2$, and (**d**) NO$_2$ concentrations over the NCP in four seasons of 2019.

For MDA8 O$_3$, high concentrations over the NCP appeared in spring (April) and summer (July) while the concentrations were lower in autumn. Because O$_3$ is a typical secondary pollutant, strong sunlight in summer is favorable for O$_3$ generation.

*3.3. Emission Contribution from Typical Petrochemical Enterprises to Regional Air Quality*

3.3.1. Spatial-Temporal Distribution in the Contributions

The distribution of air pollutant concentrations and relative contribution driven by emissions from typical petrochemical enterprises is shown in Figure 5. The $SO_2$ and $NO_2$ mainly originated from primary emissions, whereas the $PM_{2.5}$ and $O_3$ were generated from precursors such as $SO_2$, $NO_2$, and VOCs. Therefore, the surface $SO_2$ and $NO_2$ driven by emissions from the four typical petrochemical enterprises demonstrated the characteristic of near-source distribution (Figure 5c,d) while being primarily distributed near enterprise A and enterprise B with large emissions. The surface $PM_{2.5}$ driven by emissions from the four typical petrochemical enterprises demonstrated near-source characteristics while being primarily distributed near enterprise B. The surface $O_3$ driven by emissions from the four typical petrochemical enterprises did not exhibit a clear near-source distribution and was associated with complex precursors. Moreover, the precursors took some time to generate the secondary pollutant $O_3$ and migrated a certain distance through atmospheric circulation during the transformation process. Therefore, the secondary pollutant $O_3$ did not show near-source distribution (Figure 5b).

Figure 5a illustrates that high surface $PM_{2.5}$ concentrations driven by emissions from the four typical petrochemical enterprises mainly occurred in spring (April). The relative contribution to the surface $PM_{2.5}$ concentrations driven by the four typical petrochemical enterprises was significantly low in winter, which can be attributed to the high emissions from other sectors in winter.

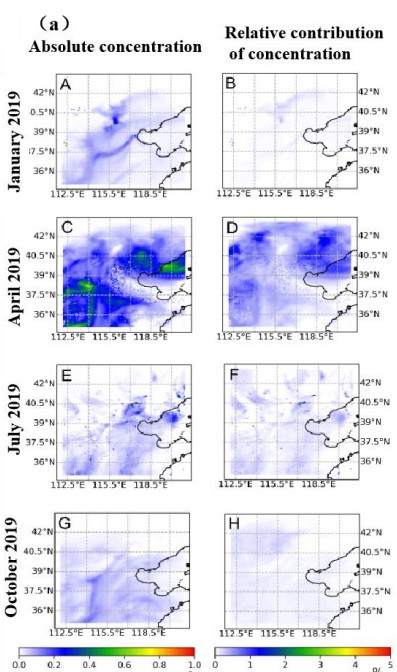

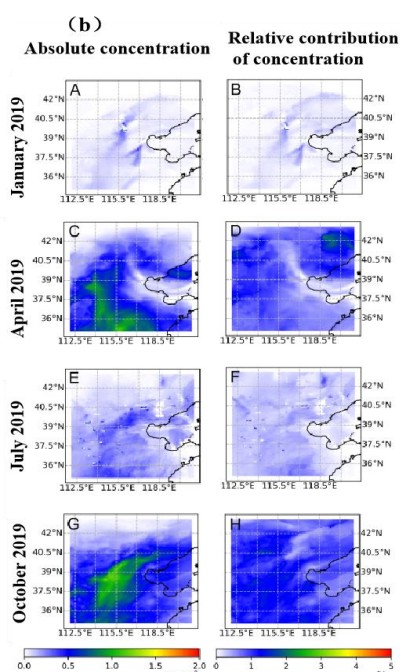

**Figure 5.** *Cont.*

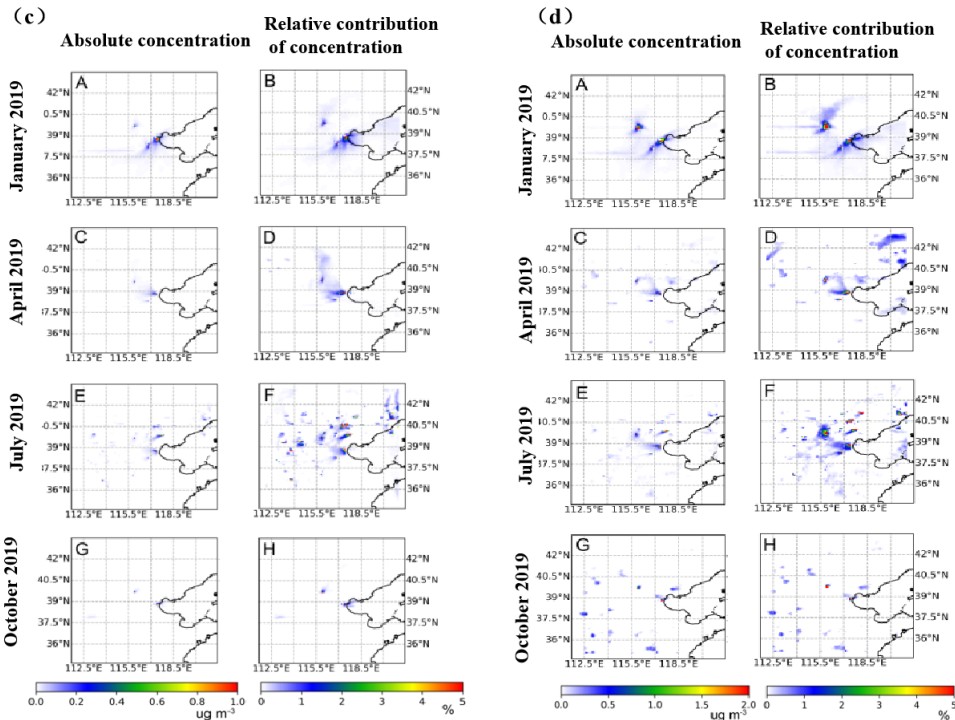

**Figure 5.** Distribution of air pollutants ((**a**) PM$_{2.5}$, (**b**) MDA8 O$_3$, (**c**) SO$_2$, (**d**) NO$_2$) concentrations and relative contribution driven by emissions from four typical petrochemical enterprises in 2019.

For MDA8 O$_3$, high surface concentrations driven by emissions from the four typical petrochemical enterprises mainly occurred in spring (April) and autumn (October), while the concentrations were low in winter (January) and summer (July). For regional distribution, the concentration hotspot in spring was mainly located in western Shandong province as well as central and Southern Hebei province, whereas the hotspot in autumn was mainly located in the NCP.

For SO$_2$ and NO$_2$, high contributions to the surface concentrations driven by emissions from four typical petrochemical enterprises mainly occurred in winter (January) and summer (July). Enterprise B in Beijing and A in Tianjin were the major contributors for the two months.

### 3.3.2. Quantitative Contributions Based on the Distance from the Sources

The emission contributions from the four typical petrochemical enterprises to regional air quality were quantified based on the distance from the sources (Table 2). Results demonstrated that the contribution of emissions from the typical petrochemical enterprises to regional air quality decreased significantly with the increase in distance, especially for SO$_2$ and NO$_2$, which showed near-source characteristics. Typical petrochemical enterprises significantly influenced regional NO$_2$ pollution. For the distance of 9 km, the contributions from enterprises A and B exceeded 20% while having the highest contribution of 46.5% (within the distance of 9 km near enterprise A in January). With the increase in distance, the contribution decreased significantly. The contributions were between 0.15 and 4.5% when the distance reached 263 km for NO$_2$.

**Table 2.** Contribution ratio (%) of surface $PM_{2.5}$, $SO_2$, and $NO_2$ concentrations driven by emissions from the four typical petrochemical enterprises within different distances from the enterprises.

| | | | <9 km | <27 km | <81 km | <243 km |
|---|---|---|---|---|---|---|
| **Enterprise A** | $PM_{2.5}$ | Jan. | 0.06 | 0.05 | 0.05 | 0.04 |
| | | Apr. | 0.30 | 0.28 | 0.28 | 0.32 |
| | | Jul. | 0.13 | 0.12 | 0.12 | 0.12 |
| | | Oct. | 0.04 | 0.05 | 0.06 | 0.07 |
| | $SO_2$ | Jan. | 2.76 | 1.17 | 0.61 | 0.28 |
| | | Apr. | 1.37 | 0.80 | 0.43 | 0.17 |
| | | Jul. | 1.03 | 0.48 | 0.26 | 0.19 |
| | | Oct. | 1.80 | 0.51 | 0.18 | 0.06 |
| | $NO_2$ | Jan. | 2.79 | 1.22 | 0.62 | 0.32 |
| | | Apr. | 2.78 | 1.28 | 0.59 | 0.26 |
| | | Jul. | 2.24 | 1.00 | 0.53 | 0.31 |
| | | Oct. | 2.93 | 0.90 | 0.31 | 0.14 |
| **Enterprise B** | $PM_{2.5}$ | Jan. | 0.18 | 0.10 | 0.07 | 0.05 |
| | | Apr. | 0.27 | 0.27 | 0.30 | 0.34 |
| | | Jul. | 0.03 | 0.04 | 0.07 | 0.12 |
| | | Oct. | 0.06 | 0.05 | 0.05 | 0.06 |
| | $SO_2$ | Jan. | 0.94 | 0.35 | 0.13 | 0.19 |
| | | Apr. | 0.70 | 0.34 | 0.16 | 0.19 |
| | | Jul. | 0.64 | 0.42 | 0.20 | 0.26 |
| | | Oct. | 1.01 | 0.31 | 0.08 | 0.07 |
| | $NO_2$ | Jan. | 4.65 | 1.65 | 0.62 | 0.36 |
| | | Apr. | 2.17 | 0.87 | 0.32 | 0.30 |
| | | Jul. | 1.64 | 1.04 | 0.50 | 0.45 |
| | | Oct. | 2.88 | 0.78 | 0.22 | 0.17 |
| **Enterprise C** | $PM_{2.5}$ | Jan. | 0.09 | 0.07 | 0.05 | 0.04 |
| | | Apr. | 0.09 | 0.11 | 0.18 | 0.25 |
| | | Jul. | 0.13 | 0.11 | 0.12 | 0.11 |
| | | Oct. | 0.07 | 0.07 | 0.07 | 0.07 |
| | $SO_2$ | Jan. | 1.05 | 0.75 | 0.80 | 0.26 |
| | | Apr. | 0.43 | 0.36 | 0.53 | 0.14 |
| | | Jul. | 0.47 | 0.41 | 0.37 | 0.12 |
| | | Oct. | 0.00 | 0.00 | 0.21 | 0.05 |
| | $NO_2$ | Jan. | 1.13 | 0.88 | 0.85 | 0.32 |
| | | Apr. | 0.49 | 0.30 | 0.72 | 0.21 |
| | | Jul. | 1.32 | 0.97 | 0.90 | 0.29 |
| | | Oct. | 0.00 | 0.00 | 0.36 | 0.10 |
| **Enterprise D** | PM2.5 | Jan. | 0.03 | 0.03 | 0.03 | 0.04 |
| | | Apr. | 0.39 | 0.41 | 0.43 | 0.42 |
| | | Jul. | 0.09 | 0.07 | 0.08 | 0.08 |
| | | Oct. | 0.10 | 0.10 | 0.10 | 0.08 |
| | $SO_2$ | Jan. | 0.05 | 0.04 | 0.03 | 0.03 |
| | | Apr. | 0.01 | 0.00 | 0.00 | 0.01 |
| | | Jul. | 0.02 | 0.01 | 0.02 | 0.03 |
| | | Oct. | 0.00 | 0.00 | 0.00 | 0.00 |
| | $NO_2$ | Jan. | 0.07 | 0.05 | 0.03 | 0.04 |
| | | Apr. | 0.05 | 0.05 | 0.04 | 0.02 |
| | | Jul. | 0.20 | 0.12 | 0.11 | 0.10 |
| | | Oct. | 0.00 | 0.00 | 0.04 | 0.02 |

Enterprise A contributed the highest to regional $SO_2$ pollution in January while reaching 2.76% within a 9 km distance. The contributions in each month decreased significantly with the increase in distance. The contribution was between 0.0% and 0.28% for the 263 km distance. There were no obvious seasonal differences in the contributions from regional $NO_2$ and $SO_2$ pollution.

However, for $PM_{2.5}$, there is no significant change in $PM_{2.5}$ concentration within different distances from the enterprises. The secondary pollutants ($PM_{2.5}$) did not show

near-source characteristics. The contributions from the typical petrochemical enterprises were relatively low, which were less than 0.5%. This can be attributed to the complex precursors and secondary chemical reactions during $PM_{2.5}$ generation. The maximum contributions driven by the typical petrochemical enterprises (A, B, and D) to regional $PM_{2.5}$ pollution occurred in April (Table 2). The maximum contribution driven by the petrochemical enterprise C to regional $PM_{2.5}$ pollution occurred in July. The contribution varied little with seasons.

Through the comparative analysis of four enterprises, we found that enterprise D had the least impact on the regional air quality, while the other three enterprises had a remarkable influence on regional air quality. The difference can be attributed to the lower emissions from enterprise D compared with other enterprises.

### 3.3.3. Quantitative Contributions to the Air Quality over the NCP Region

As an area of great concern in China, the Beijing–Tianjin–Hebei (NCP) region was chosen as a study area to determine the contribution of emissions driven by the typical petrochemical enterprises to regional air quality (Table 3). Results demonstrated that for $PM_{2.5}$, the highest seasonal variations in the contributions from the four typical petrochemical enterprises were observed for the NCP region, with the maximum in April (0.42%). However, the contributions in other months were relatively low (<0.1%). For $SO_2$, the contributions of the four typical petrochemical enterprises to the NCP region were around 0.1%. The highest contributions were observed in January and July. For $NO_2$, the contributions of the four typical petrochemical enterprises to the NCP region were around 0.2%. The highest contributions were observed in January and July, which was consistent with $SO_2$.

**Table 3.** Surface concentrations of air pollutants over the NCP region caused by emissions from the four typical petrochemical enterprises and relative contributions to total atmospheric concentrations of the air pollutants.

| Pollutants | Month | Total Atmospheric Concentrations ($\mu g\ m^{-3}$) | Concentrations Driven by Petrochemical Enterprises ($\times 10^{-3}\ \mu g\ m^{-3}$) s | Contribution Ratio (%) |
|---|---|---|---|---|
| $PM_{2.5}$ | Jan. | 129.1 | 57.7 | 0.05 |
| | Apr. | 46.1 | 193.1 | 0.42 |
| | Jul. | 34.1 | 33.6 | 0.10 |
| | Oct. | 55.0 | 39.0 | 0.07 |
| $SO_2$ | Jan. | 10.1 | 11.4 | 0.11 |
| | Apr. | 4.5 | 2.9 | 0.06 |
| | Jul. | 3.5 | 4.7 | 0.14 |
| | Oct. | 5.7 | 1.4 | 0.02 |
| $NO_2$ | Jan. | 19.0 | 36.8 | 0.19 |
| | Apr. | 9.4 | 12.0 | 0.13 |
| | Jul. | 7.5 | 18.7 | 0.25 |
| | Oct. | 13.2 | 10.9 | 0.08 |

Note: Atmospheric concentrations are the average concentrations of air pollutants within the administrative boundary of the NCP region.

## 4. Discussion

As the primary precursor of MDA8 $O_3$, VOCs play an important role through tropospheric photochemical reactions [31]. In China, the annual anthropogenic VOC emission reached 16.5–23.2 million tons in recent years, which was predominantly contributed by the petrochemical industry (17–22%). In Beijing, the annual anthropogenic VOC emission was estimated to be 236–308 kilotons in recent years and the petrochemical industry was the biggest contributor (20.3%) [32]. Comparing source structures derived from positive matrix factorization and high-resolution anthropogenic VOC emission inventory, the petrochemical industry was found to be a significant contributor to anthropogenic VOCs (24.0%) in Beihai, China [33]. After facing serious anthropogenic pollution, the Ministry of Ecology and Environment of the People's Republic of China issued the second national survey of pollution sources in 2020 [34]. According to this survey, the contribution of petrochemical enterprises to regional MDA8 $O_3$ was around 1~2%. It was clear that the petrochemical

industry significantly contributed to the regional MDA8 $O_3$ concentration. As an important precursor, VOC control is particularly important. Wei et al. [35] indicated that chemical profiles of VOCs emissions from the refinery mainly included alkanes (60.0% ± 4.3%), alkenes (21.1% ± 5.5%), and aromatics (18.9% ± 3.9%). Future control efforts should focus on controlling the VOC emission from petrochemical enterprises as they are precursor pollutants of ozone and have certain regional risks.

Petrochemical enterprises contributed less to regional $PM_{2.5}$ (basically less than 0.5%) compared with other pollutants. The precursors of $PM_{2.5}$ are mainly $SO_2$ and $NO_x$, while the precursors of ozone are $NO_x$ and VOCs. The generation of $PM_{2.5}$ and MDA8 $O_3$ takes a specific amount of time. They can migrate a certain distance through atmospheric circulation during the transformation process. Therefore, the secondary pollutants did not show near-source characteristics [36]. Owing to both natural and anthropogenic sources, $PM_{2.5}$ is a complex mixture of primary and secondary particles. Anthropogenic sources have increased $PM_{2.5}$ pollution. Industry sources were found to contribute 2~40% of $PM_{2.5}$ mass with an average of 9% in China [37–39]. A previous study showed that the coal-fired power plant increased the concentrations of $PM_{2.5}$ by ~5% in a typical polluted city located in the 2 + 26 city cluster (Beijing, Tianjin and 26 other cities) in the NCP [40]. Our study found that the largest contribution of a single petrochemical enterprise to $PM_{2.5}$ in the NCP occurred in April (0.42%) while the contribution of other months was less than 0.1%. Compared with coal-fired power plants, the contributions of petrochemical enterprises to $PM_{2.5}$ in this region were relatively low.

Based on our study, petrochemical enterprises significantly contributed to $SO_2$ and $NO_2$ (especially $NO_2$) with the contribution reaching up to 4.65%. The impact of $SO_2$ and $NO_2$ produced by petrochemical enterprises decreased significantly with the increase in distance, showing significant near-source characteristics. The contributions were around 0.1~0.2% to the NCP region. A 300 MW coal-fired power plant in China contributed 0.5% and 0.3% to $SO_2$ and $NO_2$ in Fenwei Plain, respectively [41]. In contrast, the contributions from petrochemical enterprises are relatively low. Moreover, WRF simulation is known to be sensitive to the type of atmospheric event and the combinations of physical parametrizations. Furthermore, we need to evaluate the possibilities of modeling processes, taking into account different atmospheric events and parameterizations [20].

Considering that industrial pollution is the primary source of environmental pollution in China [42], improving industrial sector regulations is an inevitable choice to effectively enhance regional air quality and triumph in the fight against pollution. The Chinese government has issued a series of strict policies and regulations to curb industrial pollution. The total number of industrial pollution control facilities in wastewater and waste gas has increased 4.7 times (79,000 in 2000 to 374,000 in 2015) [43]. Emissions of air pollutants such as $SO_2$, $NO_x$, and particulate matter from industries in China gradually decreased from 2016 to 2019 as a result of "2016–2019 national ecological environment statistical bulletin" implemented by the Ministry of Ecological Environment of China. With the continuous promotion of cleaner production, the contribution of the petrochemical industry to regional air quality will decrease in the future.

## 5. Conclusions

The WRF-GC model was selected as the basic model for this study. The environmental air quality data, meteorological data, and industrial pollutant emission data of key cities in NCP from 2017 to 2019 were used. The environmental air quality prediction results of the model were compared with the monitoring data of each environmental monitoring site. The model also demonstrated good performance on the simulations of $PM_{2.5}$, $SO_2$, and $NO_2$, especially $SO_2$ and $NO_2$.

The established model was used to simulate the effect of pollutant emissions from four typical petrochemical enterprises on regional air quality. The surface $PM_{2.5}$, $SO_2$, and $NO_2$ driven by petrochemical enterprises showed near source distribution characteristics, which were primarily located near B and A enterprises with large emissions. The surface

MDA8 O$_3$ driven by petrochemical enterprises did not show near-source distribution characteristics, which were closely related to complex precursors and secondary reactions.

The petrochemical enterprise had a small contribution to SO$_2$ and NO$_2$ while the largest contribution was up to 4.65% within the distance of 9 km from enterprise A. Petrochemical enterprises had a relatively low contribution to regional PM$_{2.5}$ (less than 0.5%) within the distance of 9 km from enterprise A. The contribution of four typical petrochemical enterprises in the NCP to local pollution decreased significantly with the increase in distance. The contributions to SO$_2$ and NO$_2$ pollution were approximately 0.1~0.2% in the NCP region, with the maximum contribution occurring in January and July, respectively. The maximum contribution to PM$_{2.5}$ (0.42%) in this region occurred in April while the contribution of other months was below 0.1%. The air pollutant emissions from the four typical petrochemical enterprises had minimal impact on the air quality in the NCP while having a significant impact on the air quality near the enterprise.

**Author Contributions:** Conceptualization, W.Y. and S.Z.; methodology, L.C. and Z.Z.; software, L.C. and Z.Z.; validation, Z.Z., W.Y. and L.C.; formal analysis, Z.Z.; investigation, W.Y. and S.Z.; resources, W.Y.; data curation, W.Y. and L.C.; writing—original draft preparation, Z.Z.; writing—review and editing, Z.Z. and L.C.; visualization, Z.Z. and L.C.; supervision, W.Y. and S.Z.; project administration, S.Z.; funding acquisition, S.Z. and L.C. All authors have read and agreed to the published version of the manuscript.

**Funding:** This research was funded by [National Natural Science Foundation of China] grant number [42077200] and The computation was funded by [ECNU Public Platform for Innovation] grant number [001].

**Institutional Review Board Statement:** Not applicable.

**Informed Consent Statement:** Not applicable.

**Data Availability Statement:** Observations derived from the Chinese national air quality monitoring network (China National Environmental Monitoring Center, http://106.37.208.233:20035 (accessed on 26 August 2020)) were used for model evaluation. The emission data are available at http://111.62.218.180:9051/appIndex/psIndex/a768f05658a041c8956b29f0dd0f89a3/dataPublic accessed on 12 March 2020 and http://111.62.218.180:9051/; https://zxjc.sthj.tj.gov.cn:8888/PollutionMonitor-tj/publish.do accessed on 12 March 2020.

**Acknowledgments:** This study was funded by the National Natural Science Foundation of China (42077200). The computation was supported by the ECNU Public Platform for Innovation (001). We thank Tzung-May Fu for helps on the WRF-GC code.

**Conflicts of Interest:** The authors declare no conflict of interest.

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
