# Peer review of "Impacts of Pollutant Emissions from Typical Petrochemical Enterprises on Air Quality in the North China Plain"

_atmosphere, doi:10.3390/atmos14030545_

Round 1
Reviewer 1 Report
Authors presented the manuscript entitled “Impacts of pollutant emissions from typical petrochemical enterprises on air quality in the North China Plain”.
We have several recommendations for this manuscript which may improve it:
- Figure 3 and 4. This figures shows the dependencies between measured and modeled characteristics. Analyzing these dependencies, we can note that the agreement between the measured and modeled data is poor. The generated point clouds are often elongated in a completely different direction. At first glance, this may indicate incorrect adaptation and use of the model. However, if we take into account that the analyzed characteristics are measured locally (at a given site) and the modeled data correspond to the large volume of the atmosphere - a fairly large cell (horizontal resolution is equal to 9 km) then we can expect a similar picture. At the same time, for individual characteristics, relationships are observed. We ask add a detailed explanation of this discrepancy in terms of model under different physical atmospheric conditions (with variable thermal stratification).
- The authors use the WRF model in their research. We recommend the authors to significantly expand the description of the used model configuration. Please pay attention to the parameterization schemes for atmospheric processes. In particular, what parametrization schemes are used to describe the atmospheric boundary layer and the surface layer of the atmosphere. Why did you choose these schemes.
- It would be useful to clarify the initial and boundary conditions in the simulation.
- Table 3. This table contains a large number of “Contribution ration (‰)” Please correct the table. I think you may delete the column with words “Contribution ration (‰)” .
- It would also be useful to indicate other latest studies with the WRF model in the introduction. Perhaps you can make some comparison. We recommend the following papers:
- Shikhovtsev AY, Kovadlo PG, Lezhenin AA, Korobov OA, Kiselev AV, Russkikh IV, Kolobov DY, Shikhovtsev MY. Influence of Atmospheric Flow Structure on Optical Turbulence Characteristics. Applied Sciences. 2023; 13(3):1282. https://doi.org/10.3390/app13031282
-Di Bernardino, A.; Mazzarella, V.; Pecci, M.; Casasanta, G.; Cacciani, M.; Ferretti, R. Interaction of the Sea Breeze with the Urban Area of Rome: WRF Meso-scale and WRF Large-Eddy Simulations Compared to Ground-Based Observations. Bound.-Layer Meteorol. 2022, 185, 333–363.
-Lysenko S., A.; Zaiko, P.O. Estimates of the Earth surface influence on the accuracy of numerical prediction of air temperature in Belarus using the WRF model. Hydrometeorol. Res. Forecast. 2021, 382, 50–68.
Author Response
Dear reviewer,
Thank you very much for your valuable suggestions and comments, which significantly contribute to the improvement of this manuscript. Responses to the comments are presented behind Response.
- General comment:Figure 3 and 4. This figures shows the dependencies between measured and modeled characteristics. Analyzing these dependencies, we can note that the agreement between the measured and modeled data is poor. The generated point clouds are often elongated in a completely different direction. At first glance, this may indicate incorrect adaptation and use of the model. However, if we take into account that the analyzed characteristics are measured locally (at a given site) and the modeled data correspond to the large volume of the atmosphere - a fairly large cell (horizontal resolution is equal to 9 km) then we can expect a similar picture. At the same time, for individual characteristics, relationships are observed. We ask add a detailed explanation of this discrepancy in terms of model under different physical atmospheric conditions (with variable thermal stratification).
Response: Thanks for the comment. We totally agree with the reviewer that the average conditions in coarse model grids sometimes cannot capture the local conditions around the monitoring site adequately, such as point emission source, local microclimate, and vertical thermal structure. In this regard, we integrated the 760 monitoring stations into 238 cities. The cities with five or more monitoring stations were involved in the evaluation. Eventually, a total of 50 cities were involved and the average of observations in each city was archived. This integration produced average observations in cities and to some extent contributed to coordination of conditions between simulations and observation. Accordingly, we involved observations from four months (only January in the previous manuscript) and re-evaluated our model performance (see the new figure in Section 3.2.1).
However, the model performance on MDA8 O3 was still slightly worse than that of the other three types of pollutants. The simulated values to some extent underestimated the observed values. This bias were also attributed to the model representation of local conditions in monitoring sites. We added a detailed explanation of this discrepancy in terms of model under different physical atmospheric conditions according to reviewer’s suggestion.
“Indeed, the monitoring site was located in a given site while the model results represented the average conditions in a region with coarse horizontal resolution (9 km × 9 km) and coarse vertical stratification (altitude of surface layer, 123 m). Owing to the representation of average conditions, the model may not capture the local conditions around the monitoring site adequately, such as point emission source, local microclimate, and vertical thermal structure. These discrepancies particularly contributed to model bias in air pollutants such as O3 which have complex chemical processes with large inputs of meteorological variables. Higher horizontal and vertical resolution with more accurate meteorological and emission inputs are needed in future studies to improve regional simulations. Nevertheless, the impacts of pollutant emissions on air quality in this study were illustrated as percentage contribution, and the bias may be reduced because bias from different scenarios may largely offset each other with normalization.”
- General comment:The authors use the WRF model in their research. We recommend the authors to significantly expand the description of the used model configuration. Please pay attention to the parameterization schemes for atmospheric processes. In particular, what parametrization schemes are used to describe the atmospheric boundary layer and the surface layer of the atmosphere. Why did you choose these schemes. It would be useful to clarify the initial and boundary conditions in the simulation.
Response: Thanks for the suggestion. We clarified the parametrization schemes in our simulation on Lines 146-150, page 4. Some schemes were default settings referred from the WRF model, and some schemes were the latest ones and most widely used.
“Physical options used in our WRF-GC simulation includes Morrison two-moment scheme (Morrison et al., 2009) for microphysics, RRTMG scheme (Iacono et al., 2008) for longwave and shortwave radiation, MM5 Monin-Obukhov scheme (Jimenez et al., 2012) for surface layer, Noah scheme (Chen and Dudhia, 2001a, b) for land surface, MYNN2 scheme (Nakanishi and Niino, 2006) for planetary boundary layer, and New Tiedtke scheme (Tiedtke, 1989) for cumulus.”
- General comment:It would also be useful to indicate other latest studies with the WRF model in the introduction. Perhaps you can make some comparison. We recommend the following papers:
- Shikhovtsev AY, Kovadlo PG, Lezhenin AA, Korobov OA, Kiselev AV, Russkikh IV, Kolobov DY, Shikhovtsev MY. Influence of Atmospheric Flow Structure on Optical Turbulence Characteristics. Applied Sciences. 2023; 13(3):1282. https://doi.org/10.3390/app13031282
-Di Bernardino, A.; Mazzarella, V.; Pecci, M.; Casasanta, G.; Cacciani, M.; Ferretti, R. Interaction of the Sea Breeze with the Urban Area of Rome: WRF Meso-scale and WRF Large-Eddy Simulations Compared to Ground-Based Observations. Bound.-Layer Meteorol. 2022, 185, 333–363.
-Lysenko S., A.; Zaiko, P.O. Estimates of the Earth surface influence on the accuracy of numerical prediction of air temperature in Belarus using the WRF model. Hydrometeorol. Res. Forecast. 2021, 382, 50–68.
Response: Thanks for the suggestion and variable references. We have referenced to the above papers in the section of model description which may be more suitable than introduction (See lines 137-141, page 4). We have added relevant papers in the results and discussion section, and put forward further revision directions according to the references. See lines 223-226, page 6; lines 376-380, page 13.
- Specificcomment:Table 3. This table contains a large number of “Contribution ration (‰)” Please correct the table. I think you may delete the column with words “Contribution ration (‰)”
Response: Thanks for the suggestion. “Contribution ration (‰)” is redundant information in Table 2 in revise manuscript and we have deleted it. We added the unit (‰) in the table title. See lines 313-314, page 9.

Reviewer 2 Report
Paper deals with very significant and actual topic of air quality and impacts of pollutant emissions from petrochemical industry. The research is additionally interesting as it's conducted in China, which is known to have major issues with air pollution. The paper is well written and in accordance with all the rules of scientific writing. Aim of the paper is clearly stated and unambiguous. Methods and data are presented clearly enough to allow repetition of the study. Results and discussion with conclusions respond to the aim of the study. Structure and statement are adequate. Pertinent literature has been cited.
Based on the above, it is recommended to accept the article for publication (after minor corrections) and categorize it as an original scientific paper.
Some suggestions for paper improvements are:
- Citations and references should be corrected according to Instructions for authors (numbered in order of appearance in text, reference numbers in square brackets...)
- In Material and methods, provide basic info on four petrochemical enterprises (basic data - size, production capacity...) - it corelates, explains different amounts of annual emissions.
- Line 159/Table 1 - check sentence "The total emission of SO2 and NOx was more than 2000 tons/year, whereas the 159 total emission of NOx and VOCs was more than 2000 tons/year"
- Line 196 - "... ,the ratios of..." - small letter
- Line 281-287/Table 3 - additionally comment no significant change (decrease with the increase of distance) in PM2.5 concentration within different distances from the enterprises (decrease)
- Line 329 - font size!
- Line 390-404 - should not be part of the text!
Author Response
Dear reviewer,
Thank you very much for your valuable suggestions and comments, which significantly contribute to the improvement of this manuscript. Responses to the comments are presented behind Response.
- General comment: Paper deals with very significant and actual topic of air quality and impacts of pollutant emissions from petrochemical industry. The research is additionally interesting as it's conducted in China, which is known to have major issues with air pollution. The paper is well written and in accordance with all the rules of scientific writing. Aim of the paper is clearly stated and unambiguous. Methods and data are presented clearly enough to allow repetition of the study. Results and discussion with conclusions respond to the aim of the study. Structure and statement are adequate. Pertinent literature has been cited.
Based on the above, it is recommended to accept the article for publication (after minor corrections) and categorize it as an original scientific paper.
Response: Thanks for your positive evaluation. We have revised the manuscript according to all your valuable suggestions.
- General comment: Citations and references should be corrected according to Instructions for authors (numbered in order of appearance in text, reference numbers in square brackets...)
Response: Thanks for the suggestion. We changed the forms of citation according to Instructions for authors.
- General comment: In Material and methods, provide basic info on four petrochemical enterprises (basic data - size, production capacity...) - it corelates, explains different amounts of annual emissions.
Response: Thanks for the suggestion. Accordingly, we have provided more basic information on the petrochemical enterprises, including the size of the company, the amount of oil refining , and the number of installations (See lines 93-101, page 2). In Part 3.1, the differences in annual emissions are also described (Table 1).
- Specificcomment:Line 159/Table 1 - check sentence "The total emission of SO2 and NOx was more than 2000 tons/year, whereas the 159 total emission of NOx and VOCs was more than 2000 tons/year"
Response: We apologize for our inadequate grammar in the original manuscript. We have revised the sentence, and check other sentences throughout the manuscript. See lines 180-181, page 4.
- Specificcomment:Line 196 - "... , the ratios of..." - small letter
Response: Thanks for the comment. We have changed upper case to lower case. See lines 305, page 10.
- Specificcomment:Line 281-287/Table 3 - additionally comment no significant change (decrease with the increase of distance) in PM2.5 concentration within different distances from the enterprises (decrease)
Response: Thanks for the suggestion. We have added a description of the changes in PM2.5 in the revised manuscript. See lines 304-306, page 10.
- Specificcomment:Line 329 - font size!
Response: Thanks for the comment. We have revised the format error and checked the same errors throughout the manuscript. See lines 353, page 12.
- Specificcomment:Line 390-404 - should not be part of the text!
Response: This part may be added by the system when the paper is submitted, and it has been removed in the revised manuscript.

Round 2
Reviewer 1 Report
-I think that annual emissions of SO2, Nox and VOCs in table 1 should be rounded with the accuracy of one sign after dot (for example, 235. 45 > 235.5 t/a)
- In the sentence (line 368) the words above “the observatory” should be deleted.